# Experimental Characterization and Numerical Simulation of Voids in CFRP Components Processed by HP-RTM

**DOI:** 10.3390/ma15155249

**Published:** 2022-07-29

**Authors:** Zhewu Chen, Liansheng Peng, Zhi Xiao

**Affiliations:** 1School of Mechanical Engineering, Hunan University of Science and Technology, Xiangtan 411201, China; pengls994@126.com; 2State Key Laboratory of Advanced Design and Manufacture for Vehicle Body, Hunan University, Changsha 410082, China

**Keywords:** CFRP, HP-RTM, permeability, flexural property

## Abstract

The long cycle of manufacturing continuous carbon fiber-reinforced composite has significantly limited its application in mass vehicle production. High-pressure resin transfer molding (HP-RTM) is the process with the ability to manufacture composites in a relatively short forming cycle (<5 min) using fast reactive resin. The present study aims to investigate the influence of HP-RTM process variables including fiber volume fraction and resin injection flow rate on void characteristics, and flexural properties of manufactured CFRP components based on experiments and numerical simulations. An ultrasonic scanning system and optical microscope were selected to analyze defects, especially void characteristics. Quasi-static bending experiments were implemented for the CFRP specimens with different void contents to find their correlation with material’s flexural properties. The results showed that there was also a close correlation between void content and the flexural strength of manufactured laminates, as the flexural strength decreased by around 8% when the void content increased by ~0.5%. In most cases, the void size was smaller than 50 μm. The number of voids substantially increased with the increase in resin injection flow rate, while the potential effect of resin injection flow rate was far greater than the effect of fiber volume fraction on void contents. To form complicated CFRP components with better mechanical performance, resin injection flow rate should be carefully decided through simulations or preliminary experiments.

## 1. Introduction

Due to lightweight requirements and the fast development of electric vehicles, composite materials have been widely used to replace metallic alloys in the automobile industry [1,2,3,4]. It has been proven that composite materials, especially continuous fiber-reinforced composites, show a significant potential to increase component performance in terms of mechanical properties, saving weight and reducing the number of vehicle components. However, traditional methods of manufacturing components using continuous fiber-reinforced composites generally require a long forming time, which is difficult to implement when taking into consideration the typical short-cycle period of 1–5 min in a large-volume vehicle production line. Resin transfer molding (RTM) is one of the infusion processes mainly used for manufacturing continuous fiber-reinforced composite structures due to its low investment and shaping flexibility [5,6]. The injection pressure of the classical RTM process is generally below 2 MPa with low speeds curing resin to wet dry 3D preform [7]. Low injection pressure leads to long mold filling and preform impregnation time. Since fast curing resin is not used considering its short operation time, the RTM process inevitably increases component curing time and manufacturing cycle (30–120 min) [8]. To satisfy the automobile high volume manufacturing cycle, a high-pressure RTM (HP-RTM) process was recently proposed and reported [8,9,10], the pressure of which can reach more than 16 MPa. This alternative approach can achieve super-fast resin injection and short curing time in the mold, all of which lead to significantly different and complex process considerations compared with the traditional RTM process [11,12].

The development of the HP-RTM process has led to an enormous amount of attention from automobile companies and related research institutes. However, only limited studies related to the HP-RTM process were reported. Li et al. [13], based on Richard’s equation and a contour function, constructed a way to predict the pressure field and the resin flow front tracking at the transient situation of the VARTM process. Moreover, by comparing the results of the numerical simulation, the experiment was proven to be correct. Zhou et al. [14] investigated the influence of fibrous structures on resin filling in the RTM process and built a two-phase flow model based on Newtonian fluid and Navier–Stokes equations. They also successfully achieved the isothermal filling stage numerical simulation by combining the FVM with the volume-of-fluid approach. Binetruy et al. [15] investigated the interactions between microscopic and macroscopic flow in woven fabrics to develop more accurate Resin transfer molding. Lavacry et al. [16] adopted a numerical simulation to predict air void formation in the resin transfer molding (RTM) process and focused on the study of void formation at the macroscopic scale, while the influence of voids on the flexural properties of the structure was not investigated. Chen et al. [17] described the HP-RTM process in detail. Chaudhari et al. [10,18] briefly introduced the influences of HP-RTM process variables including injection speed, compression gap, binder concentration, and vacuum sequence on the performance of glass fiber-reinforced composite components. Results indicated a significant effect on the process parameters on the mechanical properties of the components, especially flexural properties, while less than 4% void content was also noted in the manufactured composite laminates. They also studied the sensitivity of HP-CRTM die clearance and ply sequence on the mechanical properties of laminates. Barraza et al. [19] studied the effects of flow velocity and resin viscosity on porosity, and the relationship between porosity and mechanical properties was obtained through experiments. However, details about void features in the HP-RTM process and the effects of fiber volume fractions were not investigated in the previous studies.

Fabric wetting time in the HP-RTM process has been largely limited due to the high-speed filling cycle. Inadequate wet-out can be induced by high-speed resin flow inside a fibrous preform, and consequently can lead to compromising the mechanical properties of a composite due to various defects, such as fiber wash out, void formation, or dry spots. The previous studies of autoclave and classical RTM processes have shown that the presence of such defects within the component will affect its mechanical properties to a great extent [20,21,22]. Different sizes, shapes, and locations of the manufacturing defects can also lead to various tensile or shear failure modes [23,24,25,26]. Recently, Bodaghi et al. [27] compared the void content, size, and distribution of the parts produced by different processes such as autoclave, vacuum bag processing (VBP), and high-pressure injection resin transfer method (HP-IRTM) with an injection pressure of 0.2 MPa. Results showed that the void size distribution of autoclave and VBP manufactured composite parts followed a bell shape, while those processed by HP-IRTM featured a right-skewed distribution. HP-IRTM process together with external compaction pressure yielded a composite part with the lowest void content.

Thus, the objective of the present work is to investigate the effects of HP-RTM process variables on void characteristics and flexural properties of formed CFRP components including fiber volume fraction and resin injection flow rate. Molding experiments, including various resin injection speeds and fiber volume fractions were conducted via the HP-RTM process with a high injection pressure of around 12 MPa. The corresponding forming simulation was also carried out to analyze void content and distribution. An ultrasonic scanning system and optical microscope were used to analyze the void size, feature, and content, which were then compared with the simulation results. Quasi-static bending experiments were also implemented for the specimens extracted from the formed CFRP components to find the relationship between the void contents and the flexural properties of the CFRP material.

## 2. Materials, Equipment and Methods

### 2.1. CFRP Components Processed by HP-RTM

Feature geometry combining a U-shaped structure and a flat plane was selected for investigating the HP-RTM process, as shown in Figure 1. The FE model follows the real component geometry with a size of 450 mm × 500 mm × 40 mm, while triangular shell elements meshes of 8 mm were used to mesh the geometry. The forming mold comprised 1.8 mm and 2.0 mm uniform thickness, respectively. The experimental materials used for manufacturing CFRP components were fast cure resin AM8931 supplied by Wells Advanced Materials and plain-woven carbon fibers (CC-P400-12K) manufactured by PGTEX (Jiangsu, China).

A high-pressure RTM machine with an 8000 kN composite-specific press from KraussMaffei was employed for manufacturing the CFRP components. The RTM machine was equipped with high-pressure pumps and a self-cleaning mixing head for dosing resin and hardener, which can mix both precisely and inject this mixture with a release mixed amine agent into the mold cavity at a constant flow rate (20–200 g/s) under high pressure [28,29]. The mixing ratio of epoxy resin, hardener, and the release agent was set as 100:31:2. In order to overcome fiber washout caused by high injection pressure, fabric performing and holding mechanisms in the mold were applied. After closing the mold, a vacuum process lasting 60 s was applied. The resin was injected into the mold cavity to perform impregnation at a constant flow rate of 10 g/s, 16.6 g/s, and 20 g/s, respectively. When the mold was filled, the curing time was set for 300 s with a mold temperature of 120 °C. The detailed processing parameters are listed in Table 1.

For manufacturing CFRP components, a full factorial experimental design was employed including two levels of stack sequence (4 layers in 1.8 mm thick; 5 layers in 2 mm thick) and three levels of flow rate (10 g/s, 16.6 g/s and 20 g/s) giving 6 trials. The fiber volume fraction (*V*) was calculated by the equation as follows.
(1)$$V=\frac{N\times W_{weave}}{t\times \rho_{fiber}}$$
where $\rho_{fiber}$ is fiber density; W is weave area weight; t is the thickness of single weave layer; *N* is the number of fabric layers after preforming process. Detailed process parameters and corresponding fiber volume fraction of the component are listed in Table 2.

### 2.2. Void Characterization and Flexural Property Test

Optical microscope, C-mode scanning, and image analysis are usually used for defect measurement in terms of void size, shape, content and distribution [30,31,32]. As an optical microscope can only analyze a very small area of the section of specimen each time, a C-mode scanning system (the detection resolution is 60 × 60 μm) can give an overall evaluation of defects. In this study, both Hiwave S100 ultrasonic scanning system (S100D-15005) and optical microscope VHX-5000 (Keyence, Osaka, Japan) were employed to analyze the void, characteristics and distribution of the components. The adopted frequency of the ultrasonic scanning system is 5 MHz. Specifically, the ultrasonic scanning system was used for scanning the whole sample cut from the component, while an optical microscope was employed for analyzing and verifying results from C-mode scanning.

Samples used for void measurement and flexural tests were cut from two areas (a flat plane area and a U-shaped area) with dimensions of 15 mm × 50 mm. In total, 36 specimens were prepared from 6 manufactured CFRP components, and each area included at least three samples. The samples were sequentially used for three-point bending tests. The experimental tests were implemented using an MTS E45 105 electromechanical universal testing machine (Shenzhen Yinfei Electronic Technology Co., Ltd, Shenzhen, China) at a loading speed of 2 mm/min according to the standard of GB/T1449-2005.

### 2.3. Permeability Measurements

HP-RTM process simulation was employed to determine void distribution relating to different process variables. In a liquid composite manufacturing process, permeability is generally used to characterize infiltration properties of porous media, which is a key parameter for filling prediction and macro-scale void prediction during a forming process. Many kinds of research indicated that fabric permeability was a key factor influencing resin flow rate and preform impregnation [33,34]. Specially designed permeability test equipment was manufactured in order to test the permeability of fabric stacks with different volume fractions by flexibly changing the mold cavity gap, as shown in Figure 2a. The cavity thickness can be set between 0 and 10 mm by adjusting the screw position. Figure 2b shows the experiment schematic of constant injection pressure under the unsaturated case.

The permeability test equipment comprised fixtures, balance block, screw set, aluminum plant, steel plant, transparent glass plant, and mold base. A buffer chamber was designed at the injection port. When the liquid was injected into the cavity, it passed through the buffer chamber to prevent linear characteristics of the flow front due to injection fluctuation. In order to observe the characteristics of the flow front, the upper and lower molds were made of plexiglass. This mold could test fabric permeability with various fiber volume fractions by adjusting the cavity thickness according to a bi-directional screw mechanism. In order to accurately ensure the thickness of the cavity and the parallelism of the two sides, two high-precision dial gauges were mounted on both sides of the mold to detect the thickness variation of the cavity. Four fixtures were applied to the mold to ensure the constant spacing of the upper and lower molds while ensuring the rigidity of the mold. The variations in flow front were recorded via a CCD camera and pressure sensors were used to measure the pressure changes of the liquid during the injection process. Thereafter, the permeability could be calculated using Equation (3). According to the experimentally measured permeability, it can fit the Carman–Kozeny curve to obtain empirical constants of C and n. In the experiment, pressure sensors were employed and the viscosity of the liquid was set at 0.15 Pa·s.

## 3. Results and Discussion

### 3.1. Permeability Analysis

According to Darcy’s law [35], liquid flow under a constant flow rate can be described as follows, which is valid for one-dimensional flow.
(2)$$u_{f}=-\frac{K_{x}}{\mu \left(1-V_{f}\right)}\nabla P=\frac{Q_{in}}{wh\left(1-V_{f}\right)},$$
where $K_{X}$ fabric permeability in one direction, μ is liquid viscosity; ∇P is pressure gradient; $Q_{in}$ is inlet flow rate; w is preform width; *h* is preform thickness; $V_{f}$ is fiber volume fraction. As the pressure gradient is assumed to be a linear distribution in the mold, ∇P can be expressed as:(3)$$\nabla P=-\frac{P_{in}-P_{out}}{x_{f}},$$
where $X_{f}$ is the location of the fluid flow front, Pin is inlet pressure; Pout is outlet pressure. Substituting Equation (3) to Equation (2), the fabric permeability ($K_{X}$) is then calculated by the equation as follows.
(4)$$K_{x}=\frac{t\times \mu }{(P_{in}-P_{out})\times \left(1-V_{f}\right)}{\left(\frac{Q_{in}}{w\times h}\right)}^{2},$$

Based on the abovementioned method, the obtained permeability of multi-ply fabric at different fiber volume fractions is shown in Figure 3. In the experiment, the injection pressure of the tested liquid was $0.94{\;\times \;10}^{5}\;\mathrm{Pa}$. Each fiber volume fraction was measured twice and the average value was recorded. The test rig was set with a point injection gate. The flow behavior was set with the same order of magnitude of permeability in different directions. Its good correlation with the Carman–Kozeny theory also proved the reliability and robustness of the experimental test results.

### 3.2. Numerical Simulation of HP-RTM Process

According to Darcy’s law, simulation analysis is based on PAM-RTM software using the nonconforming finite element method. Triangular meshes divide the model in order to reduce the simulation time. The model boundary condition is set to linear injection, and the injection pressure is defined as 0.94 × 105 Pa based on the experimental pressure value. It assumes that fibers are evenly distributed and the permeability tensor in the same direction is applied. The present resin and hardener mixture density is 1.19 g/cm^3^, and the viscosity curve of the mixture is shown in Figure 4. During the mold filling process of the experiment, the resin temperature before the injection is set at 100 °C, while the mold temperature is kept at 120 °C. The viscosity curve is measured by the NDJ-1 rotary viscometer.

In the process simulation, void formation follows a dual-scale porous media impregnation mechanism [36], namely capillary force theory and void velocity theory, which could be influenced by the fiber preform permeability. When the resin injection flow velocity was at a low level (<0.001 m/s), the capillary force was greater than the viscous force, which led to the fact that the resin flow rate inside the filaments was greater than that between the fiber bundles. Thereafter, the air between the filaments was surrounded to produce macro-voids. Conversely, regarding the high level of resin flow rates (>0.001 m/s), the viscous force was greater than the capillary force so the liquid flowed primarily through the interstices of the fiber bundles, causing the air between the filaments to be enveloped and form micro-voids. The void content was assumed to be related to the liquid flow rate in the logarithmic relationship. According to the related study by Patel and Xiao et al. [37,38], the calculation equation of the void content for woven glass fiber fabric is as follows.
(5)$$V_{M}=-32.28-11.8\log\left(\frac{Ca^{*}\gamma \cos\theta }{\mu }\right)$$
(6)$$V_{m}=6.35+2.35\log\left(\frac{Ca^{*}\gamma \cos\theta }{\mu }\right)$$
where $V_{M}$ represents macro void content between tows; $V_{m}$ represents micro void content between filaments in fiber tows; *Ca** is modified capillary number; γ is liquid surface tension; θ is liquid-fiber contact angle; μ is liquid viscosity.

Regarding specific injection rate with different fiber volume fraction, the relationship between injection port pressure and injection time is shown in Figure 5. The bulking point of the curve indicated that the mold cavity was completely filled by the resin. The full pressure (when the mold was fully filled) increased with the increase of injection rate and decreased with the fiber volume fraction increase, which was mainly due to the fact that as the fiber volume fraction increased, the resistance of the fibers to the resin flow increased, resulting in an increase in the pressure at the injection port. The highest full pressure (when the mold was fully filled) with 52% fiber volume fraction was about 10 MPa when the resin flow rate was 20 g/s, which was nearly twice the lowest pressure. The same results could be found at a 57% fiber volume fraction. The filling time was reduced with the increase in resin injection rate and increased with the fiber volume fraction increase, and the longest time needed to fully fill the component was about 24 s, which was also twice the shortest fill time irrespective of fiber volume fraction. This could be also explained by the increase in fiber resistance.

One typical simulation result of void content is shown in Figure 6a. The void content around the injection port was much higher than in the rest regions. The higher injection pressure existed around the injection port, which led to a shorter impregnation time and resulted in high void content. Figure 6b shows the relationship of the void content with the resin injection flow rate. It could be seen that with the increase in resin injection rate, the void content increased from 2.25% to 3.25%. The main reason for this was that the higher resin injection flow rate induced a shorter impregnation time, while the void content showed no direct relation with the fiber volume fraction. This was also reported by Matsuzaki et al. [39]. The void content on the U shape was generally higher than the plane part, which can be partly attributed to its geometric complexity. The total void content is below 3.5%, which is in good agreement with the previous HP-IRTM results reported by Khoun et al. [8].

### 3.3. Void Characterization

As shown in Figure 7a,b, results recorded with the ultrasonic scanning system were compared with corresponding optical microscope findings. SSIS refers to the level of the echo signal, and a relatively higher level of reflection intensity (red color) indicates severe distribution of void defects. At a depth of 0.4 mm, the defects were found at the same position in the optical microscope compared with the possible defects indicated by the ultrasonic scanning system. Table 3 details the comparison of void content between simulation and experimental results. Both results show a similar trend that the void content increases with resin injection flow rate and fiber volume fraction. However, the values of the void content present a large difference. As for the 52% fiber volume fraction, when the resin flow rate increased, the void content increased from about 0.42% to 2.27%, which increased more than five times. While for 57% fiber volume fraction, the void content increased from about 0.64% to 2.79%. Therefore, it could be seen that the experiment results were smaller than the simulation results, especially at a low resin flow rate. The main reason was that the specimen is a three-dimensional structure, while the ultrasonic scanning system only scanned several layers along the thickness section, while the simulation results represented the whole void content of the specimen, so the experimental results showed a relatively lower level. In addition, we extracted the specimens from the whole components for void measurement. Even for the flat part, the void content can vary in a scale concerning different sub-regions. This is also one limitation of the present study.

The optical microscope scanning result for various defects is shown in Figure 8. As seen in Figure 8a, the void width is between 5 and 20 μm with an irregular strip shape, which is distributed close to the junction of the warp and weft roving. In Figure 8b, the void shapes are close to circles with the size between 10 and 15 μm, which are trapped inside the roving bundles. Figure 8c shows a void with a width of around 150 μm and a depth of nearly 15 μm. This type of large void with a dimension bigger than 100 μm was rarely found with most of the void sizes being below 50 μm. Bodaghi et al. [34] studied the HIRTM process and showed that the majority of the pore size in the material was more than 50 μm, whereas the study of HPRTM in this study showed that the majority of the pore size in the material was less than 50 μm, and the large pores (>100 μm) were few. The void content presented an important effect on the material’s flexural property.

### 3.4. Flexural Property

Regarding different process parameters, the correlation of the CFRP flexural strength and flexural modulus to their void content is shown in Figure 9. Void content and fiber volume fraction are factors influencing flexural property. Considering the flexural strength of the CFRP specimens, it decreases significantly with the increase of the void content, which also decreases with a higher fiber volume fraction. As for the 52% fiber volume fraction, and although not statistically proven, when the void content increased from 0.5% to 2.25%, the flexural strength decreased by about 150 MPa. The void content increased from 0.8% to 2.5% with 57% fiber volume fraction, and the flexural strength decreased by about 100 MPa. Based on the tendency lines, the flexural strength decreased around 8% when the void content increased by ~0.5%. Overall results show that the void content of specimens with 57% fiber volume fraction was always higher compared to that of the 52% ones at a specific injection speed. This indicated that a proper fiber fraction volume of the component should be designed when a CFRP component is made by the HP-RTM process. According to the flexural modulus, it seems that the effect of the void content and fiber volume fraction is not evident. The reason is that the modulus is measured at the low bending deformation stage, and the fibers are more dominant than the matrix-to-flexural modulus.

## 4. Conclusions

The present study was focused on the HP-RTM process, and investigated the effect of process variables on void characteristics and flexural property. Lu et al. [40] showed that under the injection pressure of 0.10~0.35 MPa, the pore content of glass fiber continuous felt/epoxy was 9.95–3.73%, that is, the higher the injection pressure, the smaller the pore content. The flow rate and fiber volume fraction are discussed in this paper. Both experimental and simulation results showed that a higher injection flow rate (10–20 g/s) resulted in greater void content of manufactured components. The fiber volume fraction also had a significant influence on void content, so, when the fiber volume fraction increased from 52% to 57%, the void content increased by about 0.2%. The void shapes in terms of circles or irregular strips were recorded and mainly distributed inside fiber bundles or close to bundle junctions. Bodaghi et al. [27], in their study on HIRTM, showed that the majority of the pore size in the material was more than 50 μm, whereas the study of HPRTM in this paper showed that the majority of the pore size in the material was less than 50 μm, and the large pores (>100 μm) were few. It shows that HPRTM has a better practical application effect. The void content presented an important effect on the material’s flexural property. Xu et al. [41] showed that the higher the porosity is, the worse the mechanical properties are. Based on the experimental results recorded in this work, flexural strength was reduced by about 8% when the void content increased by 0.5%. More complicated structures and process variables should be considered in the future in order to thoroughly investigate their influences on void features and mechanical properties.

HP-RTM process for composite forming has recently become popular for large-volume production of automobile components. The present study investigated the effects of HP-RTM process variables on void characteristics and flexural properties of formed CFRP components including fiber volume fraction and resin injection flow rate. The voids shape in terms of circle or irregular strips were recorded and mainly distributed inside fiber bundles or close to bundle junctions. Most of the void sizes in the components were smaller than 50 μm, and very few large voids (>100 μm) were observed. The flexural strength of the manufactured CFRP was reduced by about 8% when the void content increased by 0.5%, while flexural modulus was not substantially affected by void content. The results showed that the potential effect of resin injection flow rate is far greater than the effect of fiber volume fraction on the void contents of the manufactured CFRP. To form complicated CFRP components with better mechanical performance, resin injection flow rate should be carefully decided through simulations or preliminary experiments.

## Figures and Tables

**Figure 1 materials-15-05249-f001:**
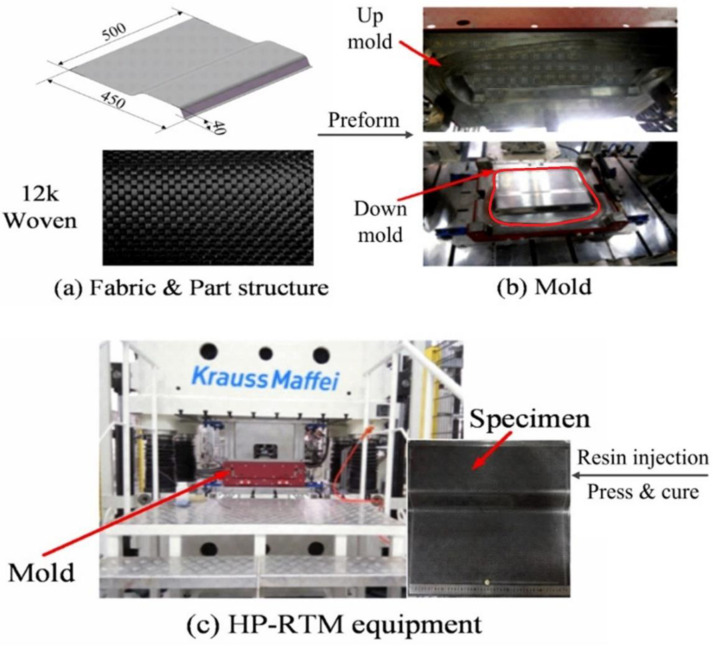
HP-RTM process for manufacturing CFRP component including (**a**) fiber fabric and part structure, (**b**) die/mold, and (**c**) HP-RTM equipment.

**Figure 2 materials-15-05249-f002:**
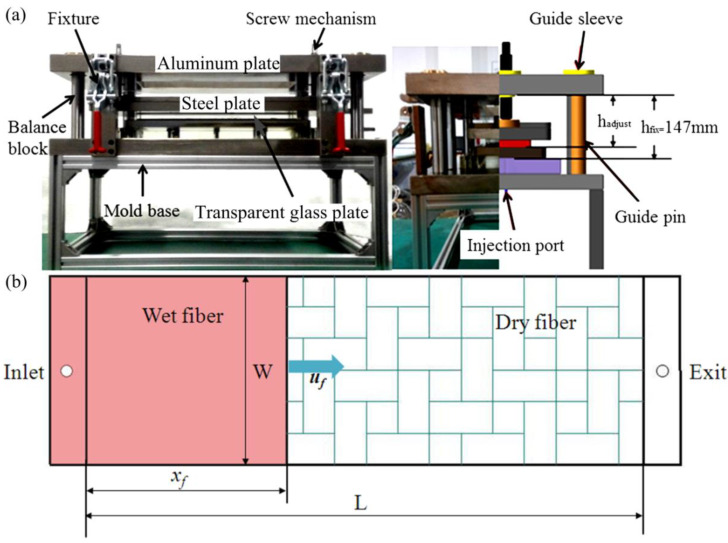
(**a**) Permeability test mold (front view and side view) and (**b**) schematic of permeability test.

**Figure 3 materials-15-05249-f003:**
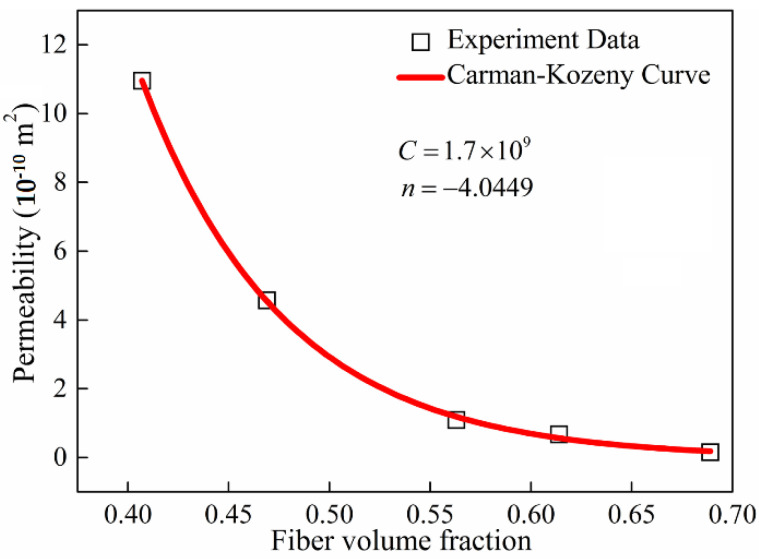
Permeability curve with varying fiber volume fractions.

**Figure 4 materials-15-05249-f004:**
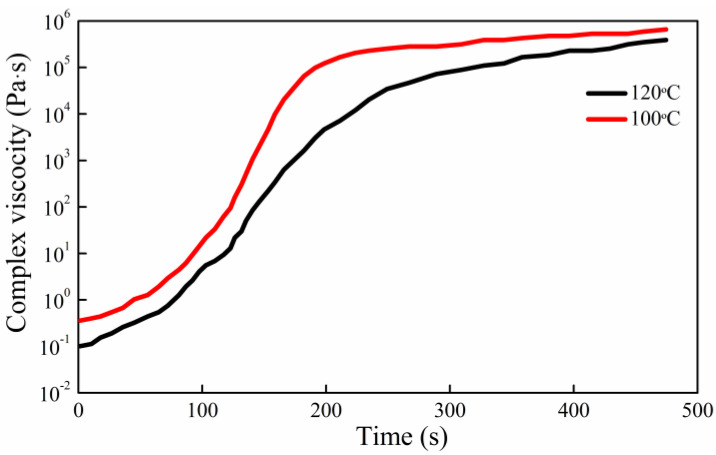
Viscosity curves of the mixed liquid at 100 °C and 120 °C.

**Figure 5 materials-15-05249-f005:**
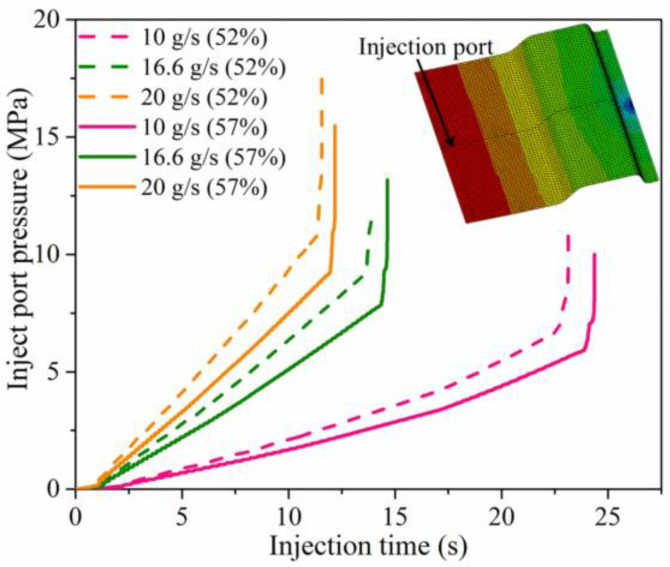
Inject port pressure against injection time curves for HP-RTM simulation.

**Figure 6 materials-15-05249-f006:**
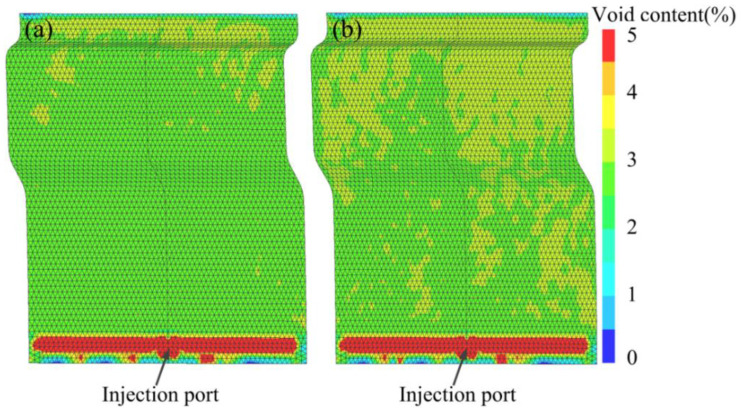
Typical simulation results showing void distribution and content with trials running with fiber volume fraction of (**a**) 52% and (**b**) 57%.

**Figure 7 materials-15-05249-f007:**
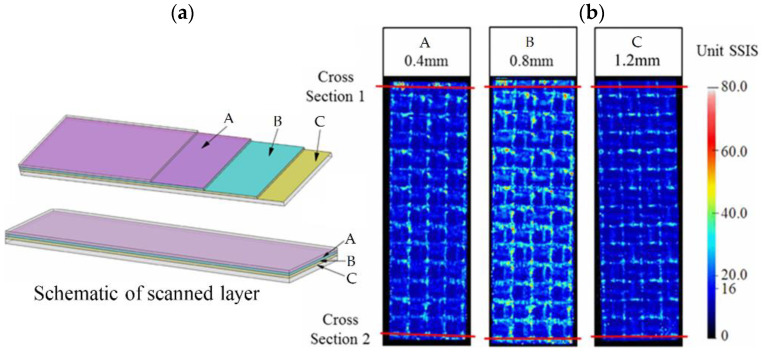
(**a**) Schematic of scanned layer; (**b**) Representative ultrasonic scanning results showing void content.

**Figure 8 materials-15-05249-f008:**
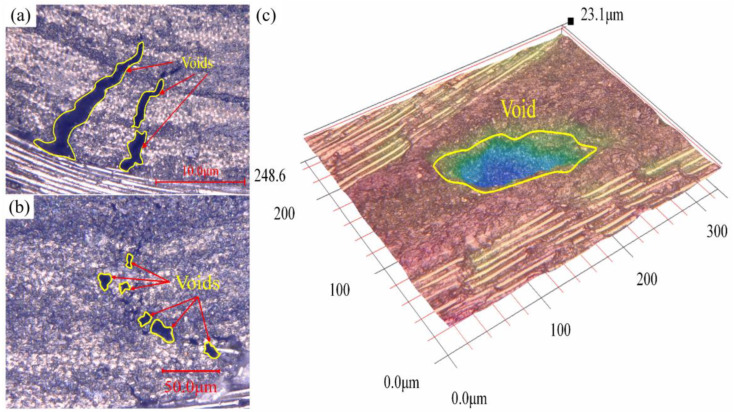
Optical microscope scanning results of the different void characteristics including (**a**) irregular strip shape, (**b**) circle shape, and (**c**) relatively large void with dimensions.

**Figure 9 materials-15-05249-f009:**
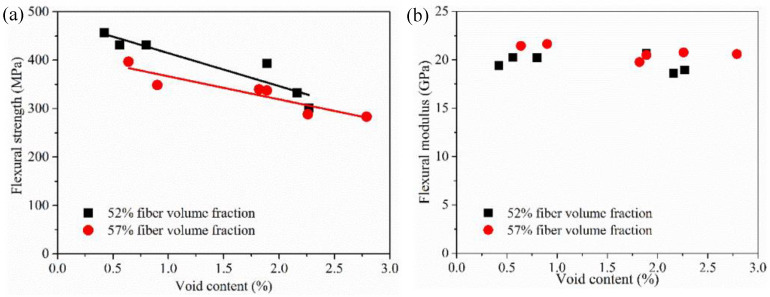
(**a**) Flexural strength and (**b**) flexural modulus with different void content and fiber volume fraction.

**Table 1 materials-15-05249-t001:** Detailed processing parameters applied for HP-RTM process.

Closing Speed	Pressing Speed	Opening Speed	Press Force	Cure Time	Resin Temperature	Hardener Temperature	Mold Temperature
800 mm/s	1–80 mm/s	800 mm/s	6000 kN	300 s	100 °C	25 °C	120 °C

**Table 2 materials-15-05249-t002:** Experimental factors and corresponding levels.

Variables	Levels		
	1	2	3
Fiber volume fraction	52%	57%	
Resin injection rate (g/s)	10	16.6	20

**Table 3 materials-15-05249-t003:** Comparison of void content between simulation and experimental results.

Test No.	Sample Location	Test Variables (Fiber Volume Fraction and Resin Injection Rate)	Simulation Result	Experimental Result
1	Flat area	57%, 10 g/s	2–2.5%	0.64%
2	U-shaped area	57%, 10 g/s	2.5–3%	0.9%
3	Flat area	57%, 16.6 g/s	2.5–3%	1.82%
4	U-shaped area	57%, 16.6 g/s	2.5–3%	1.89%
5	Flat area	57%, 20 g/s	3–3.5%	2.26%
6	U-shaped area	57%, 20 g/s	3–3.5%	2.79%
7	Flat area	52%, 10 g/s	2–2.5%	0.42%
8	U-shaped area	52%, 10 g/s	2.5–3%	0.56%
9	Flat area	52%, 16.6 g/s	2.5–3%	0.8%
10	U-shaped area	52%, 16.6 g/s	3–3.5%	1.89%
11	Flat area	52%, 20 g/s	3–3.5%	2.27%
12	U-shaped area	52%, 20 g/s	3–3.5%	2.16%

## Data Availability

Data sharing not applicable.

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
