# Peer review of "Experimental Characterization and Numerical Simulation of Voids in CFRP Components Processed by HP-RTM"

_materials, 2022, doi:10.3390/ma15155249_

Round 1

Reviewer 1 Report

This paper present a study aims to investigate the influence of HP-RTM process variables including fiber volume fraction and resin injection flow rate on void features, content and distribution of manufactured CFRP component based on experiment and numerical simulation. However the study is not correctly presented. The novelty of the work is not defined. What is new in this study versus related literature?. There is a lack of information in some sections. Figures in section 2 are not clear, and they are blurred.  What´s the preassure used in the experiments?  and what is the justification for establishing the values of the study variables? Standars are not referenced.  The numerical model is not validated (what I think is something basic when you develop a numerical model) and it is not correctly described. Fig 8(c) is missing but you metion it in the text. In section 3.3 the numbers does not match with Table 3, and I do not see the voids in Figue 8(b) clear. Dimensions and lables of the void shoulds be included in Figure 9, because I do not known how do you measure the voids. Where are those voids located? Coclusions are quite poor, I do not see clear a relevant discover. Increasing a 0.2% of void contents with a 5% increase in fibers is not really relevant. I would like to read a deeper discussion of results and more experiments that complete the results of the study.

Author Response

请参阅附件。

Reviewer 2 Report

This paper deals with a numerical and experimental studies of voids occurence during the High Pressure Resin transfer Mold (HP-RTM) process. The manuscript has to be improved taking into account the following comments :

1- The bibliography given in the introduction is too brief. There are no references on the modeling of the HP-RTM process. The novelty of the proposed research is then not very highlighted compared with the existing scientific litterature. The same work was done by Christophe Binetruy (Ecole Centrale Nantes) during the 2000's. More sophisticated numerical models were developped by Chung-Hae Park (IMT Nord Europe). Please refered to these works and show the novelty except the fact that the authors studies the HP process.

3- Concerning the experimental can the authors give more details for the C-scanning (frequency of the US transducor, post-treatment parameters) and flexural tests (number of tested specimens)?

4- The numerical resuts show no influence of the fiber volume fraction on the voids content (table 3). How the authors explain this fact? Why to choose these 2 values of fiber volume fraction (52% and 57%) which lead to  very small variation of voids content (0.2 % experimentally)?

5- Can the authors compare the obtained flexural properties with the voids content to the existing results from the scientific litterature?

2- There are many typos (missing spaces between words).

Remarks :

- Ref [5] remove [1]

- Ref [21] it is Joel Breard not Jol Breard

Reviewer 3 Report

The reviewed manuscript investigates the influence of high-pressure resin transfer molding (HP-RTM) process variables including fiber volume fraction and resin injection flow rate on void features, content, and distribution of manufactured CFRP components based on experiment and numerical simulation. The authors conducted their experiments with various resin injection speeds and fiber volume fractions via the HP-RTM process. Furthermore, the corresponding forming simulation was also conducted to analyze void content and distribution. I am sure that the study deserves the attention of readers and can be published, however, some major concerns need to be addressed before accepting the paper for publication to improve the readability and clarity of the manuscript:

1-    The use of the English language is reasonable, however, there are a number of punctuation and grammatical errors; that should be corrected and rephrased using academic English for a better flow of text for the reader. For example, lines 74, 75, 120, and 124 have some merged words that can confuse the reader. Please try to review the whole paper.

2-    Please consider reviewing the abstract and highlighting the novelty. The abstract should contain answers to some questions, what problem was studied and why is it important? Please provide a brief introduction on the applied tests used in the manuscript like flexural property testing.

3-    In the introduction, most of the references are very old. There are no references in 2022 and one reference in 2021. I think incorporating updated references show the interest of researchers in the paper topic.

4-    At the end of the introduction, it is better to give more information about evaluation tests that will be used inside the experimental work and what expected results will be extracted from such tests.

5-    The authors used a mixing ratio of epoxy resin, hardener and the release agent was set as 100:31:2. What type of agent used in the current study?  The utilized ratios are these ratios according to a standard or datasheet. Please, specify the source of the utilized ratios.

6-    The authors set the curing time 300s with mold temperature of 120 °C. Why authors selected such curing time and temperature?

7-    Samples used for void measurement and flexural tests were cut from two areas with dimensions of 15mm×50mm. The samples were sequentially used for three-point bending tests. Do the sample dimensions were selected according to a specific standard; please refer to that.

8-    In the results and discussion, Permeability analysis, the first two paragraphs, and Fig. 3 illustrate the testing process. It is better to transfer it to the Materials, equipment, and methods section.

9-    The explanation of Fig. 4 is not clear. The author said that Each fiber volume fraction was measured twice and the average value was recorded. I think the measurement should be not less than 3 times and the standard error should be calculated and added to the figure.

10-   In the Numerical simulation of the HP-RTM process, please explain the model in detail and add figures that illustrate the model.

11-   The uncertainty of the experimental measurement is not specified. Please specify the uncertainty for all measured values.  

12-    Some of the results are merely described and are limited to comparing the experimental observation and describing results. The authors are encouraged to include more detailed results and discussion sections and critically discuss the observations from this investigation with existing literature.

Please, read the text carefully before the next submission of the paper.

Round 2

Reviewer 2 Report

The manuscript have been improved. All my comments have been taking into account. The authors added only the references that I gave but it was only for information. I am not sure that the bibliography is enough exhaustive. Please for the citation do not use the first name : christophe et al. [11] and Aurelie et al. [12] have to be replaced respectively by Binetruy et al. [11] and Lebel Lavacry et al. [12]

Reviewer 3 Report

Many thanks for the revision and for incorporating all suggested changes to the manuscript that are nicely reflected. The authors did a good job to improve the article. I believe that the article has become much better but still, I feel a shortage of references (there are no references in 2022).
